In silico directed evolution of Anabas testudineus AtMP1 antimicrobial peptide to improve in vitro anticancer activity

Fazry Shazrul 1
Najm Ahmed Abdulkareem ahmadaljemeely@gmail.com 1
Mahdi Ibrahim Mahmood 2 3
Ang Arnold 2
Lee LiTing 2
Loh Choy-Theng 2 4
Syed Alwi Sharifah Sakinah 5
Li Fang 6
Law Douglas douglas.law@newinti.edu.my 2
1 Department of Food Sciences, Faculty of Science and Technology, Universiti Kebangsaan Malaysia , Bangi , Selangor , Malaysia
2 Faculty of Health and Life Sciences, INTI International University , Nilai , Negeri Sembilan , Malaysia
3 Dentistry Department, Al-Rafidain University College , Baghad , Iraq
4 Hangzhou Foreseebio Biotechnology Co., Ltd , Hangzhou , China
5 Department of Biomedical Science, Universiti Putra Malaysia , Serdang , Selangor , Malaysia
6 Jiangsu Vocational College of Medicine , Yancheng , China
Uversky Vladimir
Electronic publication date: 2024 Sep 26
Publication date: 2024
Volume: 12
Electronic Location ID: e17894
Received 2024 May 7; Accepted 2024 Jul 19
Copyright: ©2024 Fazry et al.
Copyright year: 2024
Copyright holder: Fazry et al.
License: This is an open access article distributed under the terms of the Creative Commons Attribution License, which permits unrestricted use, distribution, reproduction and adaptation in any medium and for any purpose provided that it is properly attributed. For attribution, the original author(s), title, publication source (PeerJ) and either DOI or URL of the article must be cited.
License URL: https://creativecommons.org/licenses/by/4.0/

Keywords: Anabas testudineus, Antimicrobial peptide, Breast cancer, Peptide drug, Directed evolution

Funding: The Ministry of Higher Education (MoHE), Malaysia FRGS/1/2022/STG01/INTI/02/3 FRGS/1/2023/STG01/UKM/02/5 Universiti Kebangsaan Malaysia ST-2022-013 This study was supported by the Ministry of Higher Education (MoHE), Malaysia (FRGS/1/2022/STG01/INTI/02/3) & (FRGS/1/2023/STG01/UKM/02/5), Universiti Kebangsaan Malaysia (ST-2022-013). The funders had no role in study design, data collection and analysis, decision to publish, or preparation of the manuscript.

==============================
Various studies have demonstrated that directed evolution is a powerful tool in enhancing protein properties. In this study, directed evolution was used to enhance the efficacy of synthesised Anabas testudineus AtMP1 antimicrobial peptides (AMPs) in inhibiting the proliferation of cancer cells. The modification of antimicrobial peptides (AMPs) and prediction of peptide properties using bioinformatic tools were carried out using four databases, including ADP3, CAMP-R3, AMPfun, and ANTICP. One modified antimicrobial peptide (AMP), ATMP6 (THPPTTTTTTTTTTTTTAAPARTT), was chosen based on its projected potent anticancer effect, taking into account factors such as amino acid length, net charge, anticancer activity score, and hydrophobicity. The selected AMPs were subjected to study in deep-learning databases, namely ToxIBTL and ToxinPred2, to predict their toxicity. Furthermore, the allergic properties of these antimicrobial peptides (AMPs) were verified by utilising AllerTOP and AllergenFP. Based on the results obtained from the database study, it was projected that antimicrobial peptides (AMPs) demonstrate a lack of toxicity towards human cells that is indicative of the broader population. After 48 hours of incubation, the IC50 values of ATMP6 against the HS27 and MDA-MB-231 cell lines were found to be 48.03 ± 0.013 µg/ml and 7.52 ± 0.027 µg/ml, respectively. The IC50 values of the original peptide ATMP1 against the MDA-MB-231 and HS27 cell lines were determined to be 59.6 ± 0.14 µg/ml and 8.25 ± 0.14 µg/ml, respectively, when compared. Furthermore, the results indicated that the injection of ATMP6 induced apoptosis in the MDA-MB-231 cell lines. The present investigation has revealed new opportunities for advancing novel targeted peptide therapeutics to tackle cancer.

Introduction

The documentation of systematic methodologies for the guided evolution of proteins dates back to the 1970s. The substantial substrate promiscuity exhibited by numerous proteins facilitates the recruitment of novel protein functions. The phenomenon of substrate ambiguity in a protein can be understood as the inherent capacity for evolution, enabling the protein to acquire novel specificities through mutational events or to restore its functionality through changes that deviate from the original protein sequence (Yuan et al., 2005; World Cancer Research Fund, 2021). Directed protein evolution refers to various methodologies for generating protein mutants or variants and selecting desired functional properties. In the past thirty years, guided protein evolution has become a prominent technology platform in protein engineering. The progress of this technology has been significantly enhanced by the accessibility of molecular biology tools and the emergence of high-throughput screening technologies (Yuan et al., 2005; World Cancer Research Fund, 2021; Alkatheri et al., 2022).

Antimicrobial peptides (AMPs) have recently garnered significant attention from researchers due to their ability to impede the growth of bacteria and neoplasms (Alkatheri et al., 2022; Malve, 2016; Gouic, Harnedy & FitzGerald, 2018). The categorisation of these peptides is determined by their physicochemical attributes, such as net charge, secondary structure elements, and solubility. Proteins with a comparatively low molecular weight demonstrate antibacterial and immunomodulatory characteristics against harmful microorganisms such as bacteria, viruses, and fungi. The utilisation of antimicrobial peptides (AMPs) in cancer treatment has emerged as a therapeutic approach that warrants investigation, either as a standalone intervention or combined with established conventional medicines (Ravichandran et al., 2010; Jin & Weinberg, 2019; Boparai & Sharma, 2020; Hsu, Li-Chan & Jao, 2011). AMPs can protect stem cells from microbial contamination during culture and transplantation, crucial for maintaining sterility in regenerative medicine. Additionally, certain AMPs, like LL-37, can influence stem cell differentiation, such as promoting mesenchymal stem cells to become osteoblasts, the cells responsible for bone formation. Combining AMPs with stem cell therapies can enhance tissue regeneration and provide infection protection, making this approach particularly beneficial for wound healing (Jin & Weinberg, 2019; Boparai & Sharma, 2020). This study has unveiled novel prospects for developing innovative selective peptide medicines to address cancer-related challenges.

In a recent study, the authors presented a comprehensive account of the suppressive properties exhibited by antimicrobial peptides generated from A. testudineus fish (AtMP1 and AtMP2) on the viability of MCF7 and MDA-MB-231 cells, both are breast cancer cells using apoptosis induction. This study represents the inaugural examination elucidating the underlying mechanism via which an antimicrobial peptide obtained from the mucus of A. testudineus fish induces apoptosis in MCF7 and MDA-MB-231 cells. Therefore, the current investigation utilised the MTT assay to evaluate the influence of (ATMP6) on breast cancer cell lines MCF7 and MDA-MB-231, as well as on human skin fibroblast (HS27). Furthermore, the antibacterial activity was assessed utilising the disc diffusion technique (Chen, Lin & Lin, 2009b).

Future advancements in AMP development aim to optimise and minimise the cytotoxicity towards human cells while simultaneously maximising the potential therapeutic index (Lee et al., 2003; Najm et al., 2021; Ahmed et al., 2022a; Alijani Ardeshir et al., 2020). Hence, there is a want for a novel methodology in the development of enhanced peptides. Specifically, employing directed evolution techniques to modify individual amino acid sequences of previously identified peptides would enable the discovery of more efficacious antimicrobial peptides (AMPs) that exhibit less toxicity towards normal cells. Given the findings mentioned above, it is imperative to prioritise expanding this field of study to address these challenges (Thomsen et al., 2020; Yang et al., 2013) effectively. Thus, this study aimed to create synthetic antimicrobial peptides with stronger anticancer efficacy and lower toxicity in normal cells. We employ in silico prediction methods to generate peptides and forecast their potential anticancer action and their involvement in limiting the proliferation of cancer cell lines. As a result, there is a need to investigate the modified AMPs from A. testudineus mucus and assess their cytotoxic effect against cancer cells.

Materials & Methods

Antimicrobial peptide design and alteration

The selection of the model peptide (THPPTTTTTTTTTTTTTTTAAPATTT) for this work was based on its previously demonstrated inhibitory effects on breast cancer cell lines (Chen, Lin & Lin, 2009b). A set of twenty distinct amino acids, namely alanine (A), arginine (R), asparagine (N), aspartic acid (D), cysteine (C), glutamine (Q), glutamic acid (E), glycine (G), histidine (H), leucine (L), lysine (K), methionine (M), phenylalanine (F), proline (P), serine (S), threonine (T), tryptophan (W), and tyrosine (Y), are utilised to modify the peptide. The peptides were altered by sequentially substituting each residue with other amino acids, generating new peptides with distinct sequences.

Prediction of peptide properties using bioinformatic tools

The changed peptides’ characteristics were forecasted utilising antimicrobial peptide databases, such as ADP3, CAMP-R3, and AMP fun. The peptides are analysed individually to predict antimicrobial peptides (AMPs) using ADp3. This prediction is based on the peptides’ distinctive properties, conserved structures, and amino acid composition. The forecasting process is carried out on the website (http://aps.unmc.edu/AP/). The AMPfun database, accessible at http://fdblab.csie.ncu.edu.tw/AMPfun/index.html, can predict the potential of anticancer antimicrobial peptides. By utilising the web server’s assistance, AMPfun can ascertain the activity of the AMPs. The AMPfun tool can determine the AMP status of a given peptide sequence based on its functional activity (Ahmed et al., 2022b; Wang, Li & Wang, 2016). This program employs advanced prediction algorithms to identify antimicrobial peptides (AMPs) that specifically target cancer cells and Gram-positive and Gram-negative bacteria based on their physiochemical characteristics, such as net charge, hydrophobicity, stability, molecular weight, and amino acid composition. CAMP-R3, accessible at http://www.camp.bicnirrh.res.in/, is a research tool developed to enhance and expedite antimicrobial peptide (AMP) family investigations. By leveraging the relative structural composition of antimicrobial peptides (AMPs), it can identify and produce distinct AMPs. The CAMPR3 database comprises extensive data relevant to various facets of proteins. The information mentioned above encompasses several aspects, such as sequencing, protein definition, deposit numbers, activity, parent organism, target organisms, and descriptions of certain protein families. In addition, the database offers access to external resources such as Uniport, PubMed, and other databases that specifically concentrate on antimicrobial peptides (AMPs). Consensus prediction was employed to identify potential antimicrobial peptides (AMPs) within the examined peptides. The investigation focused on the toxicity and antibacterial characteristics of these peptides (Guangshun, Xia & Wang, 2016) (Fig. 1).

ANTICP database

Each peptide’s anticancer activity was predicted using the ANTICP web server computer application (http://crdd.osdd.net/raghava/anticp/) (Ahmed et al., 2022b). SVM algorithms in databases were used to predict and characterise the potency and efficacy of AMPs with anticancer activities.

Prot param tool

The ProtParam service predicts the physical and chemical properties of proteins. The programme can be accessed using the provided Uniform Resource Locator (URL): http://web.expasy.org/protparam. The ProtParam program assessed four attributes of two cytotoxic peptides possessing a positive charge: instability, isoelectric point (PI), hydropathicity, and aliphatic index (Gasteiger et al., 2005).

Figure 1 Workflow for bioinformatics prediction of antimicrobial peptides.

Prediction of the toxicity of selected peptides

To anticipate the toxicity of protein/peptide-based therapies against human cells. The current study outlines two web-based programs, ToxinPred2 and ToxinPred3, created to forecast protein toxicity. One of the major barriers to therapeutic peptides is their toxicity to human cells, and it specially created a few existing toxicity prediction algorithms for short-length peptides (http://crdd.osdd.net/raghava/toxinpred/) (Sharma et al., 2022). The ToxIBTL dataset and associated data can be accessed without charge via the URL: https://github.com/WLYLab/ToxIBTL. ToxinPred2 is a computational methodology that predicts and designs peptides with toxic or non-toxic properties. The main dataset utilised in this methodology consists of 1,805 hazardous peptides, each consisting of 35 residues (https://github.com/WLYLab/ToxIBTL) (Wei et al., 2022).

ADME studies

The absorption, distribution, metabolism, and excretion (ADME) data for all peptides and a subset of polyphenol-peptide conjugates were obtained using the ADMETlab2.0 website (Xiong et al., 2021). Previous investigations predicted several aspects of each peptide and conjugate, including the partition coefficient, MDCK cell permeability, CYP activity, and substrate/inhibitor qualities towards Pgp.

Allergenicity prediction

The peptides’ allergenicity was predicted using the web servers AllerTOP (Dimitrov et al., 2014a) and AllergenFP (Dimitrov et al., 2014b). AllergenFP utilises a methodology that involves the application of five E-descriptor-based fingerprinting techniques. On the other hand, AllerTOP employs a predictive approach to determine the allergenicity of peptides, employing both the k-nearest neighbour (kNN) algorithm and amino acid E-descriptors.

Synthesis of the AMPS

The potential peptide for in-vitro synthesis was selected based on estimates of peptide activity obtained from an in-silico database. ATMP6 was selected based on its shown anticancer activity and net charge, as illustrated in Table 1. The peptide was synthesised and supplied by 1st BASE Co., Ltd., a company based in Singapore. Table 1 displays the sequential arrangement of ATMP6.

Cytotoxicity effect of the peptide

Cell lines

The cancer and normal cell lines exploited in this study were acquired from the American Type Culture Collection Organisation (ATCC). The cell lines employed in this investigation consisted of the MDA-MB-231 breast cancer cell line and the HS27 normal skin fibroblast cell line. The cell lines utilised in this investigation are outlined in Table 2. The culture media utilised in this investigation was DMEM (Dulbecco’s Modified Eagle media), obtained from Gibco-Thermo Fisher Scientific, Waltham, MA, USA. Gibco-Thermo Fisher Scientific provided the trypsin-EDTA solution (0.25%) used in this study. The trypsin activity unit of the solution was measured at one mmol/L. Furthermore, the Antibiotic-Antimycotic and Foetal Bovine Serum (FBS) were supplied by Gibco-Thermo Fisher Scientific. The cell lines HS27, MCF7, and MDA-MB231 were cultured in Dulbecco’s Modified Eagle Medium (DMEM) supplemented with 10% heat-inactivated foetal bovine serum (FBS), 100 units per millilitre of penicillin, and 100 micrograms per millilitre of streptomycin. The cell lines were cultured under controlled laboratory conditions, specially maintained at 37 degrees Celsius, with a carbon dioxide concentration of 5% and a humidity level of 95%.

Table 1 The sequence of ATMP6.

Peptide name	Peptide sequences	
ATMP6	THPPTTTTTTTTTTTTTAAPARTT	

Table 2 Cell lines discerption.

No.	Cell line name	Catalog no.	Description	Type of cell	Passage	
1	Hs27	CRL-2496	Homo sapiens skin, foreskin	Normal cell line	10-15	
2	MDA-MB-231	HTB-26	Homo sapiens, human	Cancer cell line	8-17	

Anticancer activity of peptides

This approach was used with a 3- (4, 5-dimethylthiazol-2-yl)-2, 5-diphenyltetrazolium bromide (MTT) test, as previously published by Najm et al. (2021) (Chen, Lin & Lin, 2009b). The media was extracted from the flask and the cells were subsequently rinsed three times in 1X phosphate buffer saline (PBS) when reaching 80% cell confluence. Following the detachment of cells from the inner surface of the flask using trypsin for 5–7 min, a volume of 5 ml of medium was introduced. The mixture was thoroughly homogenised, transferred to a fresh tube, and centrifuged at 2,500× g for 5 min. The resulting supernatant was then discarded. The cell counting procedure involved the utilisation of a hemocytometer chamber. Specifically, a cell concentration of 1 ×10 (Gouic, Harnedy & FitzGerald, 2018) was inoculated onto 96 plates that were in a healthy state. Each well of the plates contained a suspension volume of 100 µl.

The sample was aliquoted onto a well-plate in volumes of 100 µl each, with different doses of ATMP6 ranging from 1 to 10 µg. The plate was then subjected to incubation for durations of 24 and 48 h. Following a single day of incubation, a suspension was prepared by combining 50 mg of extract with 1 ml of distilled water. The assessment of cell cytotoxicity involved the dissolution of 5 mg of MTT powder in 1 ml of 1X PBS. A 10 µl of MTT reagent was administered to a cell suspension of 100 µl for 4 h. Subsequently, the contents of the wells were extracted, and a solution consisting entirely of dimethyl sulfoxide (DMSO) was introduced into the unimpaired plate for a duration of 5 to 10 min, following which the absorbance at a wavelength of 570 nm was quantified.

Determination of cell apoptosis

Annexin V-FITC apoptosis

The flow cytometry apoptosis detection kit Elabscience E-CK-A211 kit, manufactured by Elabscience firm in the United States, was utilised in this study. The cells were seeded at a density of 1 ×10 (Ravichandran et al., 2010) cells in a T25 flask, with duplicate samples for each experiment. Subsequently, the cells were treated with a 10 µg/ml concentration of ATMP6. Three T25 culture flasks were employed for control, namely for the unstained, Annexin, and propidium iodide conditions. After 48 h, the liquid portion containing apoptotic cells was gathered, and the cells adhering to the surface of each T25 culture flask, approximately 2 X 10 cells (Ravichandran et al., 2010), were subjected to trypsinisation. This process yielded a total of six falcon tubes containing a combination of both floating and trypsinised cells. The cells that were gathered underwent a washing process using phosphate-buffered saline (PBS) and were subjected to centrifugation with a force of 670 times the acceleration due to gravity (xg) for 5 min at ambient temperature. Subsequently, every pellet, consisting of approximately 2 X 10 cells (Ravichandran et al., 2010), was resuspended in 400 µl of phosphate-buffered saline (PBS). Subsequently, every pellet, consisting of approximately 2 X 10 cells (Ravichandran et al., 2010), was resuspended in 400 µl of phosphate-buffered saline (PBS). A total volume of 400 µl of cells was combined with 100 µl of incubation buffer, 2 µl of Annexin dye, and 2 µl of propidium iodide to conduct apoptosis analysis (Chen, Lin & Lin, 2009b). The results were also acquired with BD-FACSCanto II analysers.

Cell cycle profile

By this method, 1 ×106 cells/well were seeded into six-well plates and then incubated for 24 h. Following incubation, the cells received 12, 24, and 48-hour treatments with ATMP5 and ATMP6. After the cells were trypsinised and gathered, they were centrifuged at 3,000 rpm. Following collection, the pellets were fixed in 70% ethanol and stained using FxCycle RNase/PI Staining Solution (Thermo Fisher #F10797). BD-FACSCanto II analysers were utilised to read the results. The cell cycle histogram was examined using ModFit LT version 4 (Howard et al., 2021). Calculations were made to compare the cell cycle checkpoint differences to the cells that were not treated.

RNA isolation and cDNA synthesis

The process of isolating total RNA was conducted on the cancer cell lysate (MDA-MB-231), which had been generated using a cell concentration of 1 ×105. The cell lysate was purified following the manufacturer’s instructions using the AllPrep DNA/RNA Mini Kit (QIAGEN, Malaysia) (Mohammed et al., 2018; Lee & Herendeen, 1924; Yellore World Cancer Research Fund, 2021). The RNA purity and concentration assessment was conducted utilising a spectrophotometer, namely the Nanodrop 1000 manufactured by Thermo Fisher Scientific. The absorbance ratio at 260 nm to absorbance at 280 nm regularly exhibited values greater than 1.9. The cDNA synthesis procedure was performed utilising the RT∖First Strand Kit (QIAGEN Company, Malaysia) (World Cancer Research Fund, 2021) following the guidelines provided by the manufacturer. A total of 20 µl of RNA was employed as the template for this process (Ishii et al., 2019; Winkler & McGeer, 2008).

To facilitate the production of RNA for the RT (World Cancer Research Fund, 2021) profiler PCR array, the genomic DNA was combined with a reverse transcription mix consisting of a 20 µl volume. Subsequently, the specimens were subjected to an incubation phase of 15 min, exposing them to a temperature of 42 °C. The sample was incubated at 95 °C for 5 min to hinder the reaction. After that, 9 µl of the sample was diluted with 91 µl of RNA-free water and introduced into each experiment. The samples were then stored at a low temperature using ice before commencing the real-time PCR procedure. An examination of the findings was undertaken by the GeneGlobe Data Analysis Centre, which serves as a supplementary resource for real-time PCR data analysis.

Data analysis

The current investigation utilised descriptive statistics to assess quantitative data. The presented statistical measurements encompassed standard deviation (SD), mean, and percentage. The outcomes of the three distinct examinations were documented, encompassing the standard deviation (SD) and mean (mean) measurements. The descriptive percentage analysis also reported the IC50 and mucus components of the extract. A significance level of 0.05 was employed alongside a bivariate analysis to ascertain the association between the control variable and the presence of mucus or peptides. The data analysis was conducted using Microsoft Excel 2016 and SPSS version 23.

Results & Discussion

Results of peptide alteration

The bioactivity and toxicity of antimicrobial peptides (AMPs) are subject to the effect of various crucial parameters, such as the net charge, overall hydrophobicity, amino acid sequence, and amphipathicity. Efforts to improve the efficacy of synthetic antimicrobial peptides (AMPs) frequently involve manipulating one or more of these attributes, resulting in diverse degrees of success (Najm et al., 2021). The present investigation aimed to enhance and optimise the anti-cancer efficacy of AtMP1 (THPPTTTTTTTTTTTTTTTAAPATTT). This peptide had undergone prior examination and evaluation by modifying specific amino acids within the peptide. The accomplishment was attained by synthesising unique peptides employing a collection of 20 diverse amino acids. The present study involved the manipulation of individual amino acids inside the peptide to produce novel peptides, employing a collection of 20 separate amino acids. The original peptide was subjected to systematically substituting each amino acid residue, using each of the remaining 19 amino acids. The anti-cancer effectiveness of AtMP1 was further enhanced and intensified by additional investigation and assessment. A total of 428 peptides were discovered by substituting a single amino acid with 20 different amino acids. The peptides underwent analysis and selection using computational databases to identify and prioritise antimicrobial peptides (AMPs) with the highest activity level. The entire collection of peptides can be found in Supplementary File 1.

The recent study’s findings align with previous research, which showed the potential for modifying and synthesising antimicrobial peptides (AMPs) based on their three-dimensional (3D) structure. This approach aims to enhance the efficacy and selectivity of peptides while minimising their toxicity. The independent development of a predictive model for antimicrobial peptides (ACPs), hemolytic peptides, and toxic peptides was carried out utilising the three-dimensional structure of the peptides (Cheah & Yamada, 2017; Sugrani et al., 2020; E-Kobon et al., 2016; Rozek et al., 2000; Chen, Lin & Lin, 2009a).

Selection of the potential AMPs

This study employed four bioinformatics methods, including ADP3, AMPfun, CAMP-R3, and ANTICP, to forecast potential antimicrobial peptides (AMPs) derived from changed peptides. The selection of the most superior antimicrobial peptides (AMPs) was predicated upon their favourable net charge, hydrophobicity exceeding 13%, and demonstrated effectiveness against cancerous cells. The net charge criterion for AMPs is that they are typically positively charged at physiological pH due to cationic amino acids (e.g., lysine, arginine), with an optimal range of +0.0 to +9. A net charge that is too low reduces interaction with negatively charged bacterial and cancer cell membranes. At the same time, a net charge that is too high can increase hemolytic activity and potential toxicity towards mammalian cells. For overall hydrophobicity, it should facilitate the interaction of AMPs with lipid membranes but must be balanced to avoid toxicity, with an optimal range of 17% to 50% hydrophobic amino acids. Hydrophobicity that is too low results in poor membrane interaction and reduced antimicrobial and anticancer activity. However, hydrophobicity that is too high increases the likelihood of non-specific interactions and toxicity towards host cells. The amino acid sequence of AMPs determines their secondary structure, specificity, and stability. According to the ANTICP database, the optimal range for the sequence is 0 to 1 (Alijani Ardeshir et al., 2020; Thomsen et al., 2020). After evaluating the analysis outcomes conducted on the four bioinformatics databases, 212 peptides were excluded due to their lack of antimicrobial properties. Additionally, 69 peptides were eliminated as they exhibited a negative or extremely low net charge, while 48 peptides were discarded due to their low hydrophobicity. Finally, the selection process for the top ten antimicrobial peptides (AMPs) is shown in Table 3. Table 4 shows that ADP3 analysis revealed a decrease in the average hydrophobicity score of purported AMPs by 17% to 30% and a significant increase in net charge from 0.0 to +2. The peptides’ physicochemical properties, including N-terminal, C-terminal, and NC-terminal residues, were recorded in the ADP3, CAMP-R3 and AMPfun databases, leading to the selection of the top two candidates, ATMP5 and ATMP6. Table 5 shows the selection of two AMPs with the strongest anticancer capabilities based on validation using the ANTICP database. This investigation aimed to confirm the findings and identify the most effective AMPs. According to database criteria, two AMPs scored around 0.55, classifying them as anti-cancer peptides. Peptides scoring below 0.5 were deemed non-cancerous. ATMP5 and ATMP6, with scores of 0.57 and 0.59, showed the strongest anti-cancer efficacy (Thomsen et al., 2020). ATMP5 and ATMP6 underwent toxicity prediction analysis using deep-learning databases, ToxIBTL and ToxinPred2. Based on the criteria and scores from these databases, both AMPs were found to show no toxicity towards normal human cells. According to the ToxIBTL database, ATMP5 and ATMP6 had low toxicity scores of 4.1432 and 2.6253, respectively (see Table 6). In a study by Ghandehari et al. (2015), an in silico investigation was performed to evaluate the cytotoxic effects of several peptides produced from the VSVG protein on MCF-7 and MDA-MB-231 breast cancer cell lines, as well as human embryonic kidney normal cells (HEK 293) (Shahid et al., 2021). An allergenic antigen can stimulate Th2 cells, inducing B cells to generate immunoglobulin E (IgE). This IgE then attaches to FcRI, triggering the activation of eosinophils and resulting in inflammation and tissue contraction. Consequently, these processes can potentially compromise the efficacy of anticancer medications by affecting the activity of white blood cells (Dimitrov et al., 2014b). To estimate the allergenicity of our peptides, we used the online AllerTOP program, which assesses allergens using E-descriptors affined with amino acid attributes (Pierce et al., 2014). Three separate online approaches were employed to assess the level of toxicity. According to Table 7, the AllergenFP v1.0 online server shows that ATMP6 has probable allergenicity, while ATMP5 is predominantly non-allergenic. The outcomes generated by the AllerTOP v2.0 web server were consistent. The utilisation of deep learning databases, namely ToxIBTL and ToxinPred2, in predicting peptide toxicity has indicated that both antimicrobial peptides (AMPs) are deemed non-toxic towards normal human cells. The activation of Th2 cells by an allergenic antigen can produce immunoglobulin E (IgE) by B cells. This IgE then binds to FcRI, triggering the activation of eosinophils and subsequent inflammation, resulting in tissue shrinkage (Wang et al., 2016; Xie et al., 2012; Kroeze & Roth, 2012; 56).

Table 3 Details on bioinformatics prediction of antimicrobial peptides from databases.

ID	Peptide	Net charge	
seq_3541 24 bp	THPPTTTTTTTTYTTTTAAPATTT	0.25	
seq_3577 24 bp	THPPTTTTTTTTTYTTTAAPATTT	0.25	
seq_3628 24 bp	THPPTTTTTTTTTHTTTAAPATTT	0.50	
seq_3668 24 bp	THPPTTTTTTTTTTHTTAAPATTT	0.50	
seq_3690 24 bp	THPPTTTTTTTTKTTTTAAPATTT	1.25	
seq_3712 24 bp	THPPTTTTTTTTTKTTTAAPATTT	1.25	
seq_3738 24 bp	THPPTTTTTTTTTTTTTAAPATTK	1.25	
seq_3760 24 bp	THPPTTTTTTTTTTTYTAAPATTT	0.25	
seq_3785 24 bp	THPPTTTTTTTTTTTTTAAPAKTT	1.25	
seq_3833 24 bp	THPPTTTTTTTTTTTTTAAPARTT	1.25	

Table 4 Details on bioinformatics prediction of two antimicrobial peptides from databases.

Peptide sequences	ADP3	AMPFUN	CAMP R3	
	Name	Hydrophobic %	Net charge	Anti-cancer	score	Type	Score	
THPPTTTTTTTTTTTYTAAPATTT	ATMP5	13%	0.25	Yes	0.3711	AMP	0.519	
THPPTTTTTTTTTTTTTAAPARTT	ATMP6	13%	1.25	Yes	0.3659	AMP	0.512	

Table 5 Details on bioinformatic prediction of 2 antimicrobial peptides from ANTICP database.

Peptide	Score	Prediction	Hydrophobicity	Hydropathicity	Hydrophilicity	Charge	PI	
ATMP5	0.57	ANTICP	− 0.11	−0.63	0.06	0.50	7.09	
ATMP6	0.59	ANTICP	− 0.19	−0.76	0.16	1.50	10.11	

Table 6 Prediction of the toxicity of selected peptides by TOXIBTL.

Peptides	TOXIBTL	ToxinPred2	
Peptides	Prediction	Score	Prediction	Score	MW	
ATMP5	Non-toxic	4.143273e−11	Non-toxin	−0.73	2405.93	
ATMP6	Non-toxic	2.625342e−35	Non-toxic	−0.83	2433.94	

Table 7 The allergenicity profiling of the best two-peptide molecules.

Peptides	AllerTOP v. 2.0	AllergenFP v.1.0	
ID	Peptides	Prediction	Prediction	
1	ATMP5	PROBABLE ALLERGEN	PROBABLE NON-ALLERGEN	
2	ATMP6	PROBABLE ALLERGEN	PROBABLE ALLERGEN	

Results of (ADME) prediction

The analysis of pharmacokinetic characteristics of peptides using ADME studies is of utmost importance in evaluating the therapeutic potential of discovered compounds as candidates for therapy (E-Kobon et al., 2016). The utilisation of in silico ADME screening has the potential to offer valuable insights for identifying highly promising molecules, hence lowering the risk of medicine rejection (Rozek et al., 2000). Consequently, a multitude of crucial parameters were evaluated, encompassing the partition coefficient (log P), the permeability of Madin Darby canine kidney cells (MDCK), the capacity to function as a P-glycoprotein (Pgp) inhibitor or substrate, the potential to interact with cytochrome P450 (CYP) enzymes as a substrate or inhibitor, and the ability to block the human ether-a-go-go-related gene (hERG). The phenomenon of increased expression of P-glycoprotein (P-gp) and its impact on the activity of cytochrome P450 (CYP) enzymes, which serve as both substrates and inhibitors, has been observed in malignant cells. This overexpression of P-gp and altered CYP enzyme activity has been found to impede the accumulation of chemotherapeutic agents and consequently contribute to resistance against several anti-cancer drugs now in use (Chen, Lin & Lin, 2009a). The MDCK cell line has gained significant recognition as a reliable model for assessing permeability. The apparent permeability coefficient (Papp) associated with MDCK cells is commonly regarded as an indicator of the efficiency with which compounds are taken up by cells (Chen, Lin & Lin, 2009a). In addition, the developed pharmaceutical compounds mustn’t demonstrate hERG blocking, as the hERG gene, also known as the human ether-a-go-go-related gene, is critical in modulating the exchange of cardiac action potential and resting potential. Multiple studies have concluded that hERG plays a crucial role in tumour cell biology in three key activities. These functions encompass regulating cell proliferation, controlling tumour cell invasiveness using physical and functional interactions with adhesion receptors, and managing tumour cell neoplastic transformation (Kuo et al., 2018). The findings are presented in Table 8. According to our findings, in most instances, the log P values of the conjugates exhibited an increase compared to those of the individual peptides. The complete results of the ADME experiments may be found in Supplementary File 2. All the peptides exhibited negative logarithm of partition coefficient (log P) values, with ATMP5 and ATMP6 demonstrating values of −4.339 and −5.573, respectively. The compound ATMP6 exhibited the maximum permeability in MDCK cells, whereas ATMP5 had moderate permeability with values of 0.00188 and 0.0006983, respectively. The peptides lacked CYP substrates or inhibitors; however, they demonstrated the ability to generate PGP substrates in most instances. This observation suggests that this particular attribute may have an impact on drug efflux. None of the compounds elicited hERG blocking, a desirable attribute in pharmaceuticals. ToxIBTL and ToxinPred2 databases use validated algorithms to accurately predict peptide toxicity, considering essential physicochemical and structural parameters. ANTICP specialises in predicting anticancer properties with integrated data from multiple sources, improving prediction accuracy. ADP3 and CAMP-R3 focus on antimicrobial peptide analysis, employing robust algorithms validated against extensive datasets and incorporating detailed physicochemical properties like net charge and hydrophobicity for effective antimicrobial activity prediction. However, in silico predictions rely on computational models rather than direct experimental data, potentially causing discrepancies between predicted and actual outcomes by oversimplifying biological systems and neglecting critical nuances in peptide behaviour. Prediction accuracy hinges on data quality and completeness, with biased or incomplete datasets skewing results. Algorithms also assume specific peptide structure-function relationships that may not fully reflect real-world scenarios, limiting generalizability across species or conditions due to biological and environmental variations (Kroeze & Roth, 2012; 56). This study employed in vitro experiments to confirm the in silico findings.

Table 8 ADME studies results.

Compound	Log P at pH 7.4	MDCK permeability (cm/s)	Herg blocker	Pgp inhibitor/ Substrate	CYP1A2 substrate /inhibitor	
ATMP5	−4.339	0.0006983219063840806	NO	1.75/ 1.0	No / No	
ATMP6	−5.573	0.0018829464679583907	No	0.05/ 1.0	No / Yes	

Results of cytotoxic effect of AMPs

Based on the criteria to select the best peptides (Net charge, hydrophobicity, anticancer activity) and the predicted ability to interact and insert into apoptotic genes, this study selected the ATMP6 to synthesise and subsequently employed in wet laboratory verification experiments. ATMP6, which exhibited the highest level of accuracy in prediction. The MTT assay was employed to investigate the cytotoxic effects of ATMP6 on the MDA-MB-231 human Adenocarcinoma breast cancer cell line and the HS27 human skin fibroblast cell line, serving as a control. Various doses ranging from 0.625 to 20 µg/ml were administered during the experiment. According to the findings presented in (Fig. 2), The cellular viability of the MDA-MB-231 cell line exhibited a notable decrease at both the 24-hour and 48-hour time points when the concentration of peptides was augmented. The data reported in Fig. 2 indicate that the influence of ATMP5 on the HS27 cell line was negligible.

Figure 2 Cytotoxicity effect of synthetic ATMP5 on cancer MDA-MB-231 breast cancer cell line.

After 24 h, ATMP6′s IC50 values were found to be 96.20 µg/ml for HS 27 cells and 64.04 µg/ml for MDA-MB-231 cells. At 48 h, the IC50 values decreased to 48.03 ± 0.013 µg/ml for HS 27 and 7.52 ± 0.027 µg/ml for MDA-MB-231. A study by Najm et al. (2021) reported that ATMP1 had an IC50 value of 8.25 ± 0.14 µg/ml for MDA-MB-231 cells after 48 h.

Apoptosis induction by annexin V-FTIC assay

The Annexin V-FITC technique was utilised to assess the occurrence of apoptosis in MDA-MB-231 cell lines treated with ATMP6. This study aimed to evaluate the influence of ATMP6 on MDA-MB-231 cell lines through a comparative analysis of untreated and treated samples. The figure depicted in Fig. 3 provides a visual representation of the presence of early-phase apoptosis in the lower right quadrant, late-phase apoptosis in the upper right quadrant, necrotic cells in the upper left quadrant, and viable cells in the lower left quadrant. In the early stages of apoptosis, there was evidence of compromised mitochondrial membrane permeability, leading to a disrupted balance between the inner and outer layers. Phosphatidylserine was detected on the cellular membrane, and its association with Annexin V was identified. This occurrence is widely regarded as an early sign of apoptosis. During the advanced phases of apoptosis, the structural integrity of the nuclear membrane was impaired, leading to the penetration of the dye into the nucleus. Based on the data depicted in Fig. 3, it can be observed that the early apoptotic cell populations of MDA-MB-231 cells subjected to ATMP6 treatment for 48 h were recorded to be around 35.04 ± 0.32. The late apoptotic cell populations were also found to be 14.65 ± 0.24. The percentage of viable MDA-MB-231 cells not subjected to any treatment was found to be 79%, supporting the results obtained from our cell cytotoxicity experiment. In contrast to prior investigations, the study conducted by Najm et al. (2021) revealed that ATMP1 manifested an initial stage of programmed cell death in around 25% of the cellular population, while 29% of the cells were shown to be in the advanced stage of programmed cell death. Prior research has demonstrated promising results in suppressing cancer cell proliferation through antimicrobial peptides (AMPs) sourced from fish skin or mucus. Antimicrobial peptides, with antibacterial, antifungal, and antiproliferative characteristics, are abundantly present in fish mucus. A study was undertaken by E-kobon et al. to examine the cytotoxic properties of antimicrobial peptides (AMPs) extracted from the mucus of Achatina fulica on several cancer cell lines (Winkler & McGeer, 2008; Cheah & Yamada, 2017). According to a recent study, an examination was conducted to examine the inhibitory effects of the antimicrobial peptide TH2-3, obtained from tilapia skin mucus, on the growth of human fibrosarcoma (HT1080) cells. This inquiry showed a noteworthy suppression of cell growth (Wlodkowic, Skommer & Darzynkiewicz, 2009; Kuo et al., 2018; Rakers et al., 2013; Gaspar et al., 2015). Kuo et al. (2018) conducted a study showing that an antimicrobial peptide called MSP-4 induces apoptosis in osteosarcoma MG63 cells via intrinsic and extrinsic pathways. Prior studies have examined the impact of cancer drugs on the proliferation of breast cancer cells and the expression of genes linked to apoptosis (Hossam et al., 2016; Li et al., 1999; Ong et al., 2021).

Figure 3 Apoptosis detection results by Annexin V FTIC-A assay.

This assay identifies apoptotic cells based on Annexin V binding, distinguishing between early apoptosis (Annexin V positive, PI negative) and late apoptosis (both Annexin V and PI positive) using the Elabscience E-CK-A211 kit. The results were also acquired with BD-FACSCanto II analyzers.

Cell cycle analysis

Fluorescence-activated cell sorting (FACS) was utilised to count and profile cells in a heterogeneous fluid combination to perform a cell cycle study. In this experiment, the MDA-MB-231 cancer cell cycle was monitored thrice (12, 24, and 48 h) and contrasted with the untreated cell line under the same circumstances. MDA-MB-231 cell development was halted at the G0/1 phase by ATMP5 and ATMP6 (Fig. 4). When DNA damage is present, the G0/1 checkpoint prevents cancer cells from entering the S-phase, stopping the cell cycle. Moreover, as demonstrated by the earlier results in Fig. 2, ATMP5 and ATMP6 exhibited the maximum cytotoxicity (IC50).

Figure 4 The cell cycle gene expression profile in MDA-MB-231 cells.

These results meant that ATMP5 and ATMP6 could contribute to the blockage of the breast cancer cell cycle in the G0/1 phase. Several peptides and proteins have been reported to arrest the cell cycle at the G0/G1 phase and target different pathways. For example, round goby (Neogobius melanostomus) and lesser weever (Trachinus vipera) were noted to have mucus components that inhibit the growth of human cancer cells (Alijani Ardeshir et al., 2020; Fezai et al., 2016). However, existing evidence shows that the extracted compounds of Negombata magnifica do not block the growth of hepatocellular carcinoma (HCC) cell lines at the G0/G1 phase; instead, they block the HCC cell cycle at the G2/M phase (Rady et al., 2016).

Gene expression profiler RT2 PCR array results

The study employed the human apoptosis cancer RT2 Profiler PCR Array (PAHS-012ZA) to assess the apoptosis pathway in MDA-MB-231 cancer cells and identify the specific genes that regulate apoptosis in response to the experimental interventions. The present study entailed an examination of 84 pivotal genes responsible for regulating the process of programmed cell death in humans, which is popularly known as apoptosis. The genes above are linked to a specific biological pathway, including proteins (genes) that can influence several cellular processes, including cell cycle control, angiogenesis, and apoptosis. Figure 4 provides a comprehensive illustration of the alterations in gene expression induced by ATMP6 in MDA-MB-231 cancer cells over 48 h. In comparing the cells treated with ATMP5 and the control group, which included untreated cells, a selection threshold of a 1.5-fold change was employed. The information regarding fold change is succinctly illustrated in Fig. 4. During the experimental method, a comprehensive analysis was conducted to determine the differential expression of several genes. Nevertheless, it is crucial to acknowledge that the genes above had a fold change value below 1.5, necessitating meticulous scrutiny and investigation. The observed upregulation of 16 genes in the MDA-MB-231 cancer cell line was attributed to the administration of ATMP6 therapy. The genes encompassed in this set are BAD, BAX, BIK, BID, BCL10, CASP3, CASP6, CASP7, CASP8, CASP9, CASP14, RPLP0, CYC5, MCL1, TP53, and FAS. On the other hand, a decrease in the expression of BCL-2 and BCL2A1 was observed. In addition, the chemical ATMP6 exhibited a notable inhibitory impact on the advancement of the cell cycle in MDA-MB-231 breast cancer cells, particularly during the G0/1 phase. Furthermore, it triggered programmed cell death (apoptosis) in the cell above lines. The findings of this study demonstrated that the breast cancer cell line MDA-MB-231, when subjected to ATMP6 intervention, exhibited cell cycle arrest primarily in the G1 phase, initiation of apoptosis, and increased expression of the tumour suppressor gene p53. The compound ATMP6 was found to induce cellular apoptosis in MDA-MB-231 cells through the intrinsic pathway of programmed cell death. The mechanism outlined involves the activation of the p53 protein, which subsequently promotes the transcription of the pro-apoptotic gene BAX and inhibits the transcription of the anti-apoptotic gene BCL-2. The molecular modifications described above ultimately lead to the activation of specific executioner caspases, including caspase-9, caspase-3, caspase-7, and caspase-8, and the commencement of the programmed cell death process known as apoptosis. The current finding supports a previous study conducted by Najm et al. (2021) on ATMP1. The previous examination also demonstrated that ATMP1 may effectively inhibit the advancement of MDA-MB-231 cells at the G0/1 phase and induce apoptosis.

Conclusions

This research endeavour constitutes an initial investigation into the effects of antimicrobial peptides derived from the A. testudineus fish on malignant and normal cell lines. As a result, the synthesis of the ATMP6 molecule was carried out, followed by its use in in-vitro research to verify its cytotoxic effects. The current study aimed to examine the cytotoxic effects of ATMP6 on the MDA-MB-231 breast cancer cell line compared to human skin fibroblast (HS27) cells. The MDA-MB-231 cell line exhibited 75% and 24% viability after 24 and 48 h, respectively. On the other hand, the HS27 cell line exhibited 82% and 72% viabilities at 24 and 48 h, respectively. The cellular vitality of MDA-MB-231 cells was 71% and 22% at 24 and 48 h, respectively. Similarly, the study observed that the vitality of HS27 cells was determined to be 82% and 72% at the respective time points of 24 and 48 h. According to the research conducted by Najm et al. (2021), the results of this study suggest that ATMP6 reveals a significant and superior efficacy in inhibiting cancer growth when compared to ATMP1. In contrast, ATMP1 exhibits a comparably poorer ability to induce cell death in the MDA-MB-231 cell line. In brief, this study aimed to clarify the effective modification of a fish-derived antimicrobial peptide to provide improved and long-lasting anti-cancer properties while reducing harmful side effects. The efficacy of anticancer antimicrobial peptides (AMPs) in diminishing the viability of cancer cells has been demonstrated through the initiation of apoptosis. This study represents one of the initial investigations to elucidate the mechanism by which an antimicrobial peptide derived from the mucus of the A. testudineus fish induces cellular apoptosis. Consequently, the antimicrobial peptide has garnered significant attention in therapeutic medicine due to its potential implications for human disorders. Bioinformaticians systematically gather data on recently introduced peptide-based medications and antimicrobial peptides (AMPs), which are then meticulously organised into publically available databases. This study holds significant potential for advancing future investigations on the efficacy of antimicrobial peptides in inducing cancer cell death and modulating gene changes associated with apoptosis. Nevertheless, the present study has predicted the potential anticancer properties of antimicrobial peptides. However, further investigation is necessary to evaluate the impact of these peptides on other human cell lines and to characterise their specific attributes comprehensively. More investigation about AMPs and stem cells needs to be done to discover the impact of AMPs on stem cells. To conduct a comprehensive analysis of the antimicrobial peptides (AMPs) showcased in this study, further investigation employing both in vitro and in vivo models would be necessary.

Supplemental Information

Supplemental Information 1 Raw cytotoxicity data.

Supplemental Information 2 The complete list of modified peptides and their predicted properties using in silico databases (ADP3, AMPfun, and CAMP R3)

Supplemental Information 3 The complete results of the ADME in silico prediction a multitude of crucial parameters were evaluated, encompassing the partition coefficient (log P), the permeability of Madin Darby canine kidney cells (MDCK), the capacity to function as a P-glycoprotei

Additional Information and Declarations

Competing Interests

Author Contributions

Data Availability

The authors declare there are no competing interests except Choy-Theng Loh is employed by Hangzhou Foreseebio Biotechnology Co., Ltd and being appointed as adjunct Faculty Member at INTI International University. All the work done by was Choy-Theng Loh was purely academic based.

Shazrul Fazry conceived and designed the experiments, prepared figures and/or tables, and approved the final draft.

Ahmed Abdulkareem Najm conceived and designed the experiments, performed the experiments, prepared figures and/or tables, and approved the final draft.

Ibrahim Mahmood Mahdi performed the experiments, authored or reviewed drafts of the article, and approved the final draft.

Arnold Ang performed the experiments, prepared figures and/or tables, and approved the final draft.

LiTing Lee analyzed the data, prepared figures and/or tables, and approved the final draft.

Choy-Theng Loh analyzed the data, authored or reviewed drafts of the article, and approved the final draft.

Sharifah Sakinah Syed Alwi analyzed the data, authored or reviewed drafts of the article, and approved the final draft.

Fang Li analyzed the data, authored or reviewed drafts of the article, and approved the final draft.

Douglas Law conceived and designed the experiments, performed the experiments, analyzed the data, prepared figures and/or tables, and approved the final draft.

The following information was supplied regarding data availability:

Raw data are available in the Supplemental Files.

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
