# Peer review of "In silico directed evolution of Anabas testudineus AtMP1 antimicrobial peptide to improve in vitro anticancer activity"

_PeerJ, doi:10.7717/peerj.17894_

## Round 0.1 · original submission · Major Revisions

Please address concerns of both reviewers and amend your manuscript accordingly

Reviewer 1 ·

Basic reporting

This study employs directed evolution to enhance the efficacy of synthesized Anabas testudineus AtMP1 antimicrobial peptides (AMPs) in inhibiting cancer cell proliferation. The research demonstrates a systematic approach involving bioinformatics predictions, computational studies, deep-learning databases, and experimental validations. While the study presents promising results, several questions and concerns need to be addressed to ensure the robustness and clarity of the research.

Experimental design

1. ATMP6 was chosen based on factors such as amino acid length, net charge, anticancer activity score, and hydrophobicity. Could you provide more detailed criteria and the specific thresholds used for each factor in selecting ATMP6?
2. Several tools (PepFold3, RCSB PDB, Discovery Studio, HPEPDOCK, and ZDOCK) were used for Protein-Peptide docking analysis. Can you elaborate on the criteria for selecting these tools and how their results were integrated or compared to validate the interactions with genes involved in cancer management?

Validity of the findings

no comment

Additional comments

3. Given the significant role of intestinal stem cells (ISC) in maintaining gut health and their potential interaction with antimicrobial peptides, also Paneth cells are adjacent to ISC in the niche to produce antimicrobial peptides and various signaling factors, which regulate ISC stemness and function. Thus, were any studies conducted to evaluate the impact of ATMP6 on intestinal stem cells? If not, is this aspect being considered for future investigations?

Please include content related to antimicrobial peptide and stem cells in the discussion to expand the scope and evaluation of antimicrobial peptide in the article.

Reviewer 2 ·

Basic reporting

The language of the manuscript can benefit from grammatical corrections and restructuring for better clarity. For example:
"the prediction of bioinformatics" is largely used in this manuscript, which should be written as "prediction of peptide properties using bioinformatic tools"
line 294 not necessary to list all 20 amino acid names, they are prior knowledge for the readers
line 419 basically copy and paste all IC50 results from the abstract, consider rewrite in a different way


The manuscript provides an adequate introduction and background, referencing relevant prior literature to position the study within the broader context of antimicrobial peptides (AMPs) and their therapeutic potential against cancer.

The structure follows a standard format with sections for the introduction, materials and methods, results, discussion, and conclusion.

Figures and tables are relevant and well-described. However, the resolution of some figures could be improved for better clarity.
Figure3 has no legend or description.
Figure4 font too small
Consider use consistent color for all figures

Raw data is shared in the supplementary files, adhering to the journal's data-sharing policy.

The manuscript is self-contained, presenting necessary data and results to support the hypotheses tested.

Experimental design

Methods are described in sufficient detail to allow replication. The use of bioinformatics tools and databases is well-documented, and the in vitro experimental procedures are clearly outlined.

Validity of the findings

The data provided is robust, statistically sound, and well-controlled. The statistical analyses are appropriate and support the conclusions drawn. The conclusions are well-stated, directly linked to the research question, and supported by the results. Claims of causative relationships are backed by controlled experiments.

The manuscript mentions the use of various databases for predicting peptide properties, but it does not clearly explain how the final selection of ATMP6 was made from the pool of peptides. Need to provide a more detailed explanation of the criteria and process used for selecting ATMP6 from the generated peptide variants.

Discuss the limitations of in silico predictions and provide a rationale for why the chosen databases and algorithms are reliable for this study.

Additional comments

Citation format is not consistent, for example, line 305 Zhao et al, consider adding reference

---

## Round 0.2 · accepted · Accept

All concerns of the reviewers were adequately addressed and revised manuscript is acceptable now.

Reviewer 1 ·

Basic reporting

Authors have tried their best to answer my questions. I think that most of my concerns have been well addressed in this revision

Experimental design

Authors have tried their best to answer my questions. I think that most of my concerns have been well addressed in this revision

Validity of the findings

Authors have tried their best to answer my questions. I think that most of my concerns have been well addressed in this revision

Additional comments

None